# WHAT YOU PAINT IS WHAT YOU GET

## ABSTRACT

The two most prominent approaches for building adversary-resilient image classification models are adversarial training and input transformations. Despite significant advancements, adversarial training approaches struggle to generalize to unseen attacks, and the effectiveness of input transformations diminishes fast in the face of large perturbations. In general, there is a large space for improving the inherent trade-off between the accuracy and robustness of adversary-resilient models. Painting algorithms, which have not been used in adversarial training pipelines so far, capture core visual elements of images and offer a potential solution to the challenges faced by current defenses. This paper reveals a correlation between the magnitude of perturbations and the granularity of the painting process required to maximize the classification accuracy. We leverage this correlation in the proposed *Painter-CLassifier-Decisioner (PCLD)* framework, which employs adversarial training to build an ensemble of classifiers applied to a sequence of paintings with varying detalization. Benchmarks using provable adaptive attack techniques demonstrate the favorable performance of PCLD compared to state-of-the-art defenses, balancing accuracy and robustness while generalizing to unseen attacks. It extends robustness against substantial perturbations in high-resolution settings across various white-box attack methods under $\ell_\infty$-norm constraints.

## 1 INTRODUCTION

Deep learning models excel in image classification, yet they are still vulnerable to adversarial manipulation (Nguyen et al., 2015; Szegedy et al., 2014; Biggio et al., 2013). Through carefully crafted perturbations, attackers can manipulate the representations learned by models, leading to incorrect predictions (Goodfellow Ian J., 2014). These vulnerabilities present significant security concerns, especially given the integration of these models into critical domains such as autonomous driving, healthcare, and finance (Dong et al., 2020), emphasizing the disparity between current machine learning algorithms and human-level capabilities (Geirhos et al., 2018).

Two core defense approaches have been developed to address these challenges: (1) *Adversarial training* – incorporation of adversarial examples into the training data (Madry et al., 2018; Papernot et al., 2018; Zhang et al., 2019; Singh et al., 2024) – is the most successful defense approach to date. However, it faces challenges in generalizing to unseen attacks (Bai et al., 2021) and maintaining performance on benign images (Zhang et al., 2019). (2) *Defensive transformations* – transformation of the input from adversarial space to benign space by filtering out adversarial perturbations. Some defensive transformations provide theoretical guarantees (Cohen et al., 2019; Salman et al., 2020; Nie et al., 2022). Certain defensive transformation techniques have been shown to "obfuscate gradients", leading to a false sense of adversarial robustness (Buckman et al., 2018; Ma et al., 2018; Guo et al., 2018) while being vulnerable to adaptive attack strategies (Athalye et al., 2018). In addition to these approaches, detection-based defenses (Carlini & Wagner, 2017) offer complementary strategies by identifying adversarial inputs. Overall, there is an inherent trade-off between the robustness and accuracy of adversary resilient classifiers (Zhang et al., 2019), with input transformations particularly vulnerable to large perturbations.

In this paper, we combine adversarial training with a new defensive transformation technique utilizing stroke-based painting. Our key insight is that painting strokes filter out adversarial perturbations while progressively reconstructing the most important image features. As depicted in Figure 1, early coarse strokes filter out larger perturbations (better robustness) but display only the major elements of an image (lower accuracy). Later fine strokes display more image elements (better accuracy)

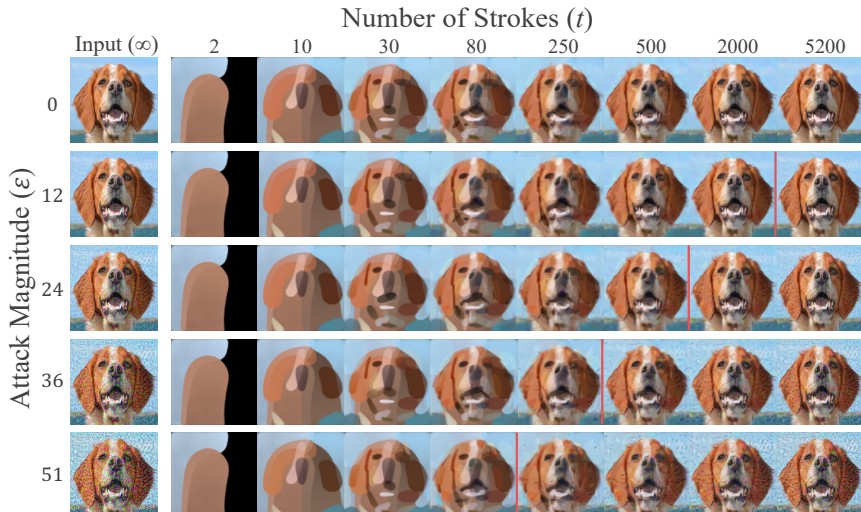

Figure 1: Painting vs adversarial perturbations. The left column includes input photos, followed rightward by their respective paintings with $t$ strokes. The top row includes benign images, followed downward by their attacked variants. Here the dog can be identified after 30 strokes. The vertical red bar marks the number of strokes when perturbations become visually perceptible. The greater the $\epsilon$, the earlier the perturbations become perceptible.

while also being affected by smaller adversarial perturbations (lower robustness). Given an image with adversarial perturbations, there is an optimal number of painting strokes – a sweet spot – maximizing the likelihood of correct classification. In realistic settings where the magnitude of the perturbations is unknown, the final decision is made based on intermediate paintings by an adversarially trained component we call a decisioner. To the best of our knowledge, this is the first attempt to integrate intermediate painting, i.e. filtering, stages into an adversarial training dataset.

We refer to the entire framework as *Painter-CLassifier-Decisioner (PCLD)*, reflecting the three core components of the proposed approach. We evaluate PCLD using state-of-the-art adaptive white-box attacks under $\ell_\infty$-norm on a subset of the ImageNet data (Deng et al., 2009). PCLD effectively extends robustness against large perturbations, generalizes to unseen attacks, and retains performance on benign images.

The most important contributions of this study are:

1. We introduce Painter-CLassifier-Decisioner (PCLD), a novel framework for adversarial training combined with stroke-based painting (Section 2.2).

2. We reveal the correlations between the granularity of the paintings and the magnitude of the attack (Section 4.3). Our findings lay the groundwork for future exploration of painting algorithms for adversarial resilience.

3. PCLD addresses the generalization challenge. Initially trained only on FGSM (Goodfellow Ian J., 2014) samples, PCLD demonstrates enhanced performance against powerful white-box attacks, including PGD (Madry et al., 2018), C&W (Carlini, 2017) and AutoAttack (Croce & Hein, 2020) (Section 4.5).

The rest of the article is structured as follows: We overview the PCLD framework and describe the main ideas and architectural highlights in Section 2. Then we discuss literature most related to PCLD in Section 3 while focusing on stroke-based defensive transformations and adversarial training techniques against which we benchmark PCLD. Section 4 presents the empirical study of PCLD as the main part of the paper. We deep dive into painting as a defensive transformation technique and assess the accuracy vs. robustness tradeoff as a function of the painting steps in Section 4.3. In Section 4.4, we assess the contribution of the adversarially trained decisioner. Section 4.5 presents the results of the PCLD benchmark against the prior art. Section 5 concludes the article.

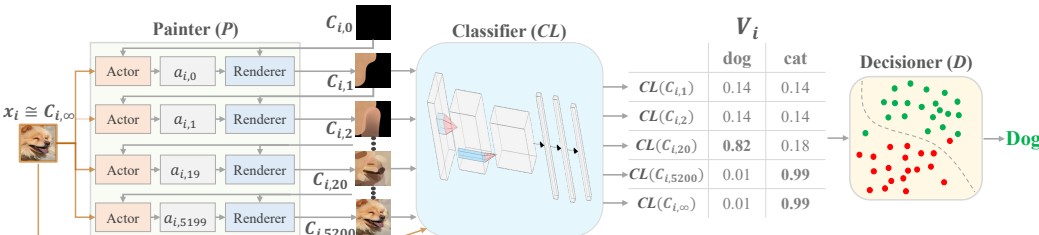

Figure 2: The overall defense framework. An input image $x_i$ benign or adversarial is processed by the Painter (P). Based on the canvas state $C_{i,t}$ and $x_i$, the actor outputs the stroke parameters $a_{i,t}$. The renderer then uses $a_{i,t}$ to render a stroke on the canvas, producing $C_{i,t+1}$. A selected set derived from the resulting canvases, together with the original input, is fed to the Classifier (CL), generating the probability instance $V_i$, which is provided to the Decisioner (D) to predict the class.

## 2 PAINTER-CLASSIFIER-DECISIONER (PCLD)

### 2.1 HIGH-LEVEL OVERVIEW OF THE PROPOSED APPROACH

In his book, Nicolaïdes (1941) emphasizes the concept of "Correct Observation" as crucial for artists to truly connect with and deeply understand the visual elements they paint. Painting algorithms (Collomosse & Hall, 2005; Li et al., 2020; Zou et al., 2021; Huang et al., 2019) attempt to capture the essential visual elements of an image. Intuitively, *focusing on the essential visual elements helps to filter out malicious perturbations while maximizing classification accuracy.*

To support this intuition, we utilize a *Painter (P)* (Huang et al., 2019) as a defense against adversarial attacks. Figure 1 shows a sequence of painting steps $t$, generated by the painter for a given input image, denoted as $t = \infty$ (left column). Among the inputs, the top is the benign image, followed downward by its adversarial variants. While the object is recognizable after a few (30-80) strokes, the perturbations are visually apparent towards the end of the drawing process, allowing early recognition of key elements for classification and avoiding malicious artifacts.

Intuitively, increasing the number of strokes ($t$) to create a more detailed painting not only clarifies the image but also reconstructs the perturbations. When the attack magnitude ($\epsilon$) is greater, the perturbations are reconstructed earlier. This demonstrates a dependence between the *Classifier's (CL)* confidence in the correct class, the painting's granularity, and the magnitude of the attack. We empirically validate this phenomenon in Section 4.3.

In real-world scenarios, however, the magnitude of an attack is unknown to the defender, making it challenging to determine the optimal step at which the painting should be provided to the classifier for decision-making. To navigate this uncertainty, a sequence of CL derived from intermediate painting steps is processed by the *Decisioner (D)*. This last component is designed to discover patterns in confidence levels, effectively addressing the challenges posed by varying attack magnitudes and the need for image clarity.

### 2.2 PAINTER DYNAMICS AND PROCEDURES

As the Painter (P) we use a pre-trained model, provided by Huang et al. (2019), which does not require retraining for new datasets. As illustrated in Figure 2 (left), the painting process begins with a target image $x_i \cong C_{i,\infty}$ and an empty canvas $C_{i,0}$. The painter decomposes the image into a sequence of strokes $a_{i,0}, a_{i,1}, ..., a_{i,n-1}$. The next stroke $a_{i,t+1}$ is derived from $x_i$ and the preceding canvas $C_{i,t}$. Rendering $a_{i,t}$ on $C_{i,t}$ generates $C_{i,t+1}$. The painter's goal is to generate the final canvas $C_{i,n}$ that is visually close to $x_i$.

For the sake of completeness, we briefly overview the employed painting process. Huang et al. (2019) model painting as a Markov Decision Process using a state space $S$, an action space $A$, a transition function $T(s_{i,t}, a_{i,t})$ and a reward function $r(s_{i,t}, a_{i,t})$. The state space contains three elements such that the current state includes: $s_{i,t} = (C_{i,t}, x_i, t)$. $s_{i,t+1} = T(s_{i,t}, a_{i,t})$ is the transition between states that results in a new stroke on the canvas. The action $a_{i,t}$, determined under a deterministic action policy, controls the position, shape, color, and transparency of the resulting stroke at

step $t$. The reward function estimates the difference between the current canvas state $C_{i,t}$ and the target image $x_i$: $r(s_{i,t}, a_{i,t}) = L_{i,t} - L_{i,t+1}$, where $L_{i,t}$ is the predicted loss by the discriminator between $x_i$ and $C_{i,t}$ and $L_{i,t+1}$ is the loss between $x_i$ and $C_{i,t+1}$. In each state, the agent's objective is to maximize the cumulative rewards in all episodes $R_{i,t} = \sum_{j=t}^{T} \gamma^{(j-t)} r(s_{i,j}, a_{i,j})$ using a decaying discounted factor $\gamma \in [0, 1]$.

*By following this approach, the painter prioritizes regenerating the essential elements of an image before recreating the adversarial perturbations.*

### 2.3 Classifier

The classifier can be any model that produces an inference vector for a given image. The most common models for this task are convolutional neural networks (CNNs). As shown in Figure 2 (middle), given a selected resulting canvas $C_{i,t}$ produced by the painter, the classifier outputs an inference vector $CL(C_{i,t})$ that contains a likelihood value for each class $c$. The matrix $V_i[t, c]$ contains the classification confidence for all classes ($c$) and the selected painting steps ($t$). The painting process contains many steps corresponding to the strokes generated by the painter. It does not make sense to apply a classifier after each stroke due to high computational costs (inference time reported in Section 4.5, Table 1).

### 2.4 Decisioner

We designed PCLD as an ensemble of classifiers that can be applied to different stages of painting the same image. Provided the classification likelihoods ($V_i$), the Decisioner (D) is responsible for making the final decision for the predicted class. Consider, for example, the likelihood in Figure 2 (right). During the initial steps, we expect the confidence in the correct class (*dog*) to increase to some point. Later, the adversarial perturbations are reconstructed by the painter, and we expect the confidence of the correct class to drop in favor of another class (*cat* in this example). A decisioner trained on adversarial examples of confidence matrices $V$ should learn to identify such patterns and select the right class. We consider two decisioner architectures: a convolutional network and a fully connected network.

## 3 Related Work

### 3.1 Stroke-Based Defensive Transformation

Kabilan et al. (2021) were the first to use sketching strokes as a defense in their framework, named VectorDefense. Given an input bitmap image $x$, VectorDefence uses the Potrace algorithm (Selinger, 2003) to transform it into a Scalable Vector Graphics (SVG) image (Ferraiolo et al., 2000) using strokes shaped from simple geometric primitives. The resulting SVG is then rasterized back into bitmap format before it feeds to the classifier. Potrace algorithm consists of 4 steps: (1) trace a given bitmap to paths by generating boundaries that divide black and white regions, (2) approximate each path by an optimal polygon, (3) smooth out each polygon, and (4) optimize the generated curve by connecting successive segments of the Bézier curve if possible. Although VectorDefense showed promise as an effective input transformation defense, it was initially evaluated solely on the MNIST dataset. In this paper, we extend the VectorDefense testing to a subset of the ImageNet dataset, which comprises more complex and high-dimensional images.

### 3.2 Adversarial Training

There have been significant advances in adversarial training over the past decade, beginning with Szegedy et al.(Szegedy et al., 2013) method of training on both adversarial and clean samples. Goodfellow et al.(Goodfellow et al., 2014) introduced an approach to generate adversarial examples by tweaking the input based on the gradient of the loss function. This was extended by Madry et al. (Madry et al., 2018) showing robustness improvements through min-max optimization. Further advancements included exploration of adversarial training on large datasets like ImageNet and strategies to counter overfitting and label leaking, such as avoiding the use of ground-truth labels (Kurakin et al., 2016).

Additional methods like Ensemble Adversarial Training (EAT) (Tramèr et al., 2017) and Unsupervised Adversarial Training (UAT) (Alayrac et al., 2019) were developed to enhance robustness against diverse adversarial attacks. Diffusion models have also been explored for adversarial training by iteratively removing adversarial noise and training on the resulting images (Wang et al., 2023). Furthermore, Vision Transformers (ViT) have been utilized in adversarial training, benefiting from their global self-attention mechanisms to enhance robustness (Singh et al., 2024).

Recent techniques have addressed the accuracy-robustness trade-off. Randomized Adversarial Training via Taylor Expansion (RATE) (Jin et al., 2023) integrates randomness during training to improve generalization, leveraging insights from both TRADES and AWP. A notable advancement named TRADES (Zhang et al., 2019) defined a theoretically principled approach to balance the benign accuracy and robustness of the model to adversarial attacks, setting a benchmark for subsequent research in this domain. TRADES won the NIPS 2018 Adversarial Vision Challenge out of 1995 defenses, marking a significant milestone in the field. It is widely used as a competitor for adversarial training methods to this day due to its robust theoretical foundations and empirical performance. We empirically compare PCLD with TRADES and with RATE combined with TRADES (denoted as Rand TRADES).

## 4 EMPIRICAL STUDY OF PCLD

As described in Section 2, our dual-layered defense strategy designed to protect the target classifier consists of two key components: (1) a series of input transformations executed by a painter and (2) post-processing of the classifier's outputs using a decisioner. In this section, we evaluate the impact of the Painter-CLassifer (PCL) model, emphasizing the critical role of the decisioner. Finally, we assess the resilience of the complete model, PCLD, and compare its performance with benchmark methods.

### 4.1 EXPERIMENTAL ENVIRONMENT

We use a balanced subset of ImageNet, comprising 7000 images of seven animals: elephant, squirrel, chicken, spider, dog, butterfly, and cat. The dataset is divided into 70% for training, 10% for validation, and the remaining 20% for testing. The original image size of 375x375 pixels was resized to 300x300 pixels and scaled in a range of 0-1. During the training phase, we incorporate random image rotations of $45°$ and apply horizontal flipping with a probability of 50%. We use the CleverHans (Papernot et al., 2018) and ART (Nicolae et al., 2018) libraries to attack the models under the $\ell_\infty$ norm. The hyperparameters of the PGD attack include a step size of $\epsilon/N_{iter}$, while default values were used for step sizes, random restarts, and confidence (for CW) in all other attacks. Computations required GPU cores; we run it on Amazon EC2 (Services, 2023) g5.24xlarge instances, including four 24GB NVIDIA-A10G GPU cores with 384GB memory. For the classifier (CL), we employ a ResNet-18 architecture, pre-trained on ImageNet, and adapted for seven classes. We select 15 distinct painting steps for all experiments: 50, 100, 200, 300, 400, 500, 600, 700, 950, 1200, 1700, 2200, 3200, 4200, 5200.

### 4.2 ADAPTIVE ATTACK STRATEGY

The painting process is convoluted with multiple iterative steps that cause "Exploding & Vanishing Gradients" by incorporating "multiple iterations of neural network evaluation, feeding the output of one computation as the input of the next" (Athalye et al., 2018). Consequently, we extend the Backward Pass Differentiable Approximation (BPDA) + Expectation Over Transformation (EOT) (Athalye et al., 2018) and substitute the painter during the backward pass while keeping the forward pass unchanged.

Each of the above paint steps approximated with an auto-encoder (e.g., for step 50, we fine-tune an encoder-decoder to mimic the painter's output at step 50 given the input image). Specifically, we use pre-trained ResNet-18 classifier, trained on ImageNet, as the encoder. We modify the ResNet18 by keeping the layers up to the third residual block. This results in a feature map of size 256 channels, which is used as the encoded representation.

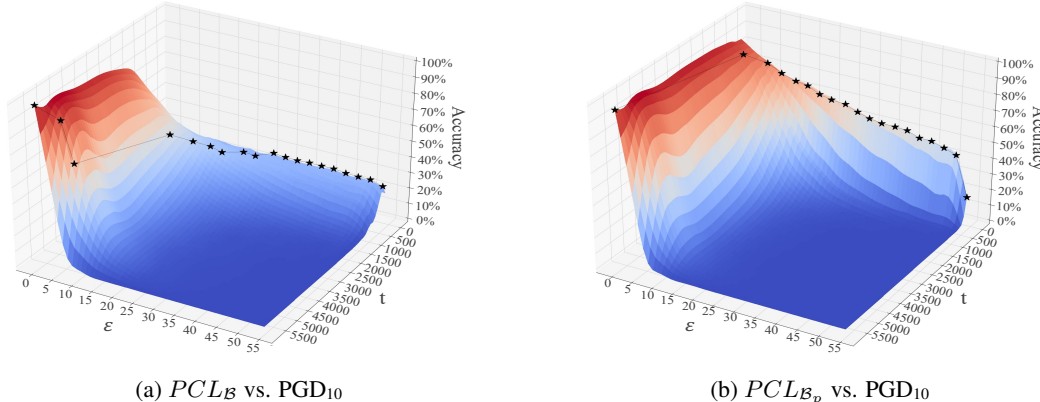

(a) $PCL_\mathcal{B}$ vs. PGD$_{10}$          (b) $PCL_{\mathcal{B}_p}$ vs. PGD$_{10}$

Figure 3: PCL test accuracy (z-axis) as a function of epsilon (x-axis, $\epsilon = 128$ scaled to $\epsilon = 54$) and paint step (y-axis, $t = \infty$ scaled to $t = 5700$). (a) $PCL_\mathcal{B}$ model - $CL_\mathcal{B}$ classifier trained only on benign images ($\mathcal{B}$). (b) $PCL_{\mathcal{B}_p}$ model - $CL_{\mathcal{B}_p}$ classifier trained on benign images and their paints. The black markers on the surface plot highlight the coordinates where the accuracy reaches peaks. The greater the attack magnitude, the earlier the painting steps in which the classifier reaches the accuracy peak.

To construct the decoder, we append a series of transposed convolutional layers to progressively upsample the encoded features back to the original image size (3x300x300). The decoder consists of four transposed convolutional layers, followed by a final convolutional layer to adjust the output to the desired 3-channel (RGB) format. We use ReLU activations for the upsampling layers and a sigmoid activation in the final layer to produce the pixel values in the range [0, 1].

Finally, the model is trained using mean squared error (MSE) as the loss function, with the Adam optimizer and a learning rate of 0.001. This approach results in 15 encoder-decoder models that replace the painter during the backward pass.

To assess the quality of our attack strategy, we perform the following sanity tests advised in (Carlini et al., 2019):

1. Compare it with a simpler naïve strategy, crafting examples using only components other than the painter, i.e., CLassifier ($CL$) for $PCL$ and CLassifier-Decisioner ($CLD$) for $PCLD$).
2. Verify that increasing the perturbation budget increases the attack success rate.
3. Generate adversarial samples with the $\epsilon = 128/255$ budget. The robustness is expected to be around random chance, as the adversary should have the ability to make any single image into a solid gray picture.
4. Verify that iterative attacks perform better than single-step attacks.

### 4.3 PAINTER-CLASSIFIER (PCL) MODEL

#### 4.3.1 TRAINING CLASSIFIER WITH PAINTS

We evaluate two training strategies for the classifier, (1) train on the benign images indicated by $\mathcal{B} = \{(x_i, y_i)\}$, and (2) train on the benign images and their paints indicated by $\mathcal{B}_p = \{(C_{i,t}, y_i)\}$. Here, $C_{i,t}$ represents the canvas in the painting step $t$ for input $x_i$, where $C_{i,\infty} = x_i$. The steps we choose to generate $\mathcal{B}_p$ start at 50 and increase to 200 in increments of 50, allowing us to closely monitor the initial significant transformations in the image's objects. Beyond 200, the increments expand to 500, continuing up to 5200. Finally, the original image $x_i$ is included, resulting in a total of 15 canvases. This method enables effective tracking of the more gradual changes as the painting process progresses. After training the classifiers $CL_\mathcal{B}$ and $CL_{\mathcal{B}_p}$, we obtain two victim models, $PCL_\mathcal{B}$ and $PCL_{\mathcal{B}_p}$, respectively. These classifiers trained using cross-entropy loss and the SGD optimizer with a learning rate of 0.01, employing a learning rate scheduler (StepLR) that reduces the learning rate by a factor of 0.1 every 7 steps.

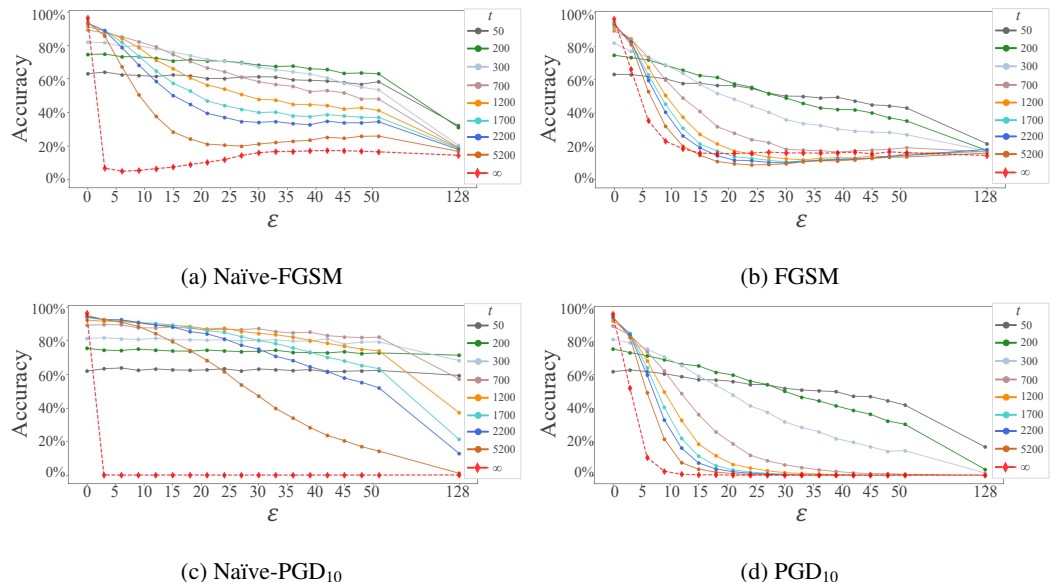

Figure 4: Test Results for attacking $PCL_{\mathcal{B}p}$. Our adaptive strategy outperforms the naive approach, especially with PGD$_{10}$ attacks. Note that when $\epsilon = 128/255$, with the naive approach, most of the PCL inferences reach high accuracy, while this anomaly is corrected by the adaptive strategy. Furthermore, with the naive strategy, FGSM yields better results than PGD$_{10}$, while the opposite is true for the adaptive strategy. This suggests that the naive strategy may discard crucial gradient information that is retained in our adaptive strategy.

To pick the best classifier training configuration, we conduct an attack on the $PCL$ model as outlined in Section 4.2. During the forward pass, the painter processes the input, while in the backward pass, the gradients associated with the loss across all painting steps are approximated using surrogate signals. The resulting gradient tensor, $g \in \mathbb{R}^{T \times W \times H \times C}$, is then averaged over the painting steps dimension $T$, producing gradients that match the shape of the input $x \in \mathbb{R}^{W \times H \times C}$. These gradients are subsequently utilized by the chosen attack algorithm.

We use a targeted PGD attack with 10 iterations on top of the above adaptive technique to attack and evaluate $PCL$ through all the selected painting steps described in Section 4.2. Figure 3 shows the test accuracy results of two $PCL$ models as a function of $\epsilon$ and $t$. Consequently, training the classifier with paints generated from the benign dataset ($\mathcal{B}_p$) seems to acknowledge more robustness to $PCL$ than training it on the benign dataset ($\mathcal{B}$) alone (data augmentation in general can improve robustness (Rebuffi et al., 2021; Addepalli et al., 2022; Li & Spratling, 2023)).

### 4.3.2 ADAPTIVE ATTACK - SANITY CHECK

Figure 4 presents the performance of $PCL_{\mathcal{B}_p}$ using both the naive strategy (left column) and the adaptive strategy (right column). It is notable that the adaptive strategy significantly improves the success rate compared to the naive method. Furthermore, achieving an accuracy significantly higher than 15% (7 classes), as shown in Figures 4a and 4c, is impossible, indicating that the gradient information used by the naive strategy is deficient. This issue is addressed by the adaptive strategy, suggesting that it effectively utilizes valuable gradients from the painting process to attack the model. Furthermore, while FGSM performs better than PGD$_{10}$ under the naive strategy, the adaptive strategy corrects this anomaly. This change is expected since multistep gradient descent methods should typically outperform single-step methods.

### 4.3.3 PERFORMANCE DYNAMICS

The relationship between the granularity of the painting, model accuracy, and the magnitude of the attack is particularly evident in Figure 3. As epsilon increases, the model reaches its accuracy peak (denoted by the black markers on the surfaces, which we refer to as $t^*$) at earlier painting steps,

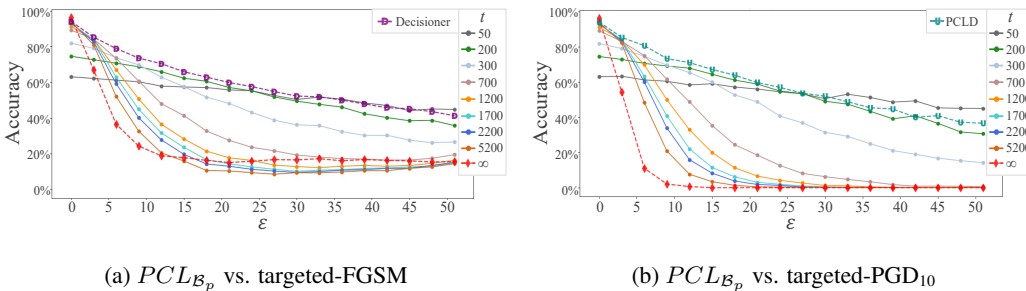

(a) $PCL_{\mathcal{B}_p}$ vs. targeted-FGSM

(b) $PCL_{\mathcal{B}_p}$ vs. targeted-PGD$_{10}$

Figure 5: Test results for attacking $PCL_{\mathcal{B}_p}$, accuracy as a function of magnitude size. The dashed line with the "D" sign in (a) is the performance of the decisioner $D_{V\Leftarrow}^{FC}$ before any attack is introduced. Likewise, the dashed line with the "U" sign in (b) is our complete model $PCL_{\mathcal{B}_p}D_{V\Leftarrow}^{FC}$ test performance for adaptive targeted-PGD$_{10}$ attacks. Although the decisioner trained only on FGSM signals, it generalizes well to PGD$_{10}$

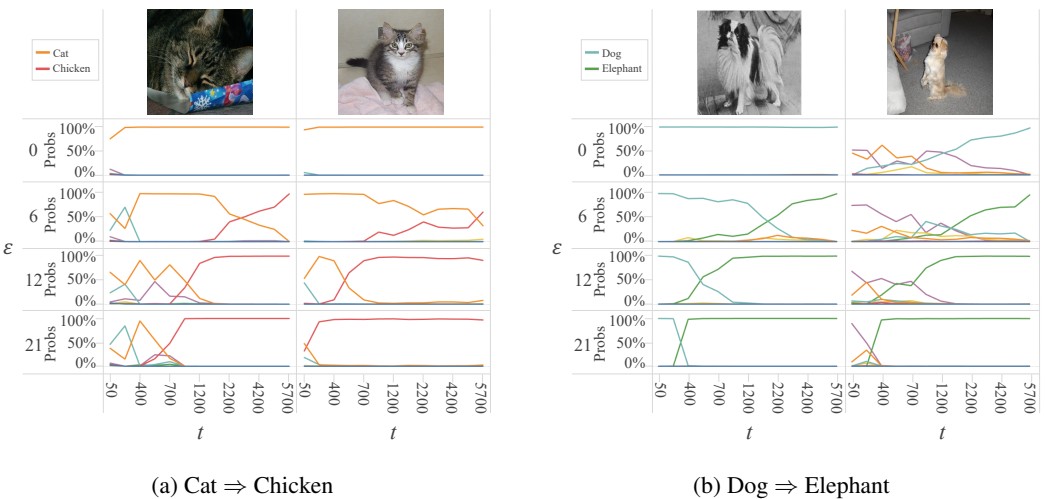

(a) Cat $\Rightarrow$ Chicken

(b) Dog $\Rightarrow$ Elephant

Figure 6: Softmax probabilities of $PCL_{\mathcal{B}_p}$ vs. targeted PGD$_{10}$ for different epsilon budgets as a function of paint-step ($t$). (a) Attack directed Cat $\Rightarrow$ Chicken. (b) Attack directed dog $\Rightarrow$ Elephant. As the perturbation increases, the shift in the highest class probability from the correct class to the targeted class occurs in earlier steps, illustrating the dynamics of confidence variation with increasing $\epsilon$.

indicating that perturbations are reconstructed sooner with a larger attack radius. We compute the Spearman correlation between $\epsilon$ and the corresponding $t$ at which the model assigns the highest probability to the correct class, denoted as $t_{prob}^*$, for each of the attacked models in Figures 3a and 3b. This correlation yields $PCL_{\mathcal{B}} : r_s(\epsilon, t_{prob}^*) = -0.54$ and $PCL_{\mathcal{B}_p} : r_s(\epsilon, t_{prob}^*) = -0.55$, both with $p \ll 0.05$. Focusing on samples where the model correctly predicted the class (i.e., the model assigned the highest probability to the correct class), the correlation coefficients strengthen to $PCL_{\mathcal{B}} : r_s(\epsilon, t_{prob}^*) = -0.76$ and $PCL_{\mathcal{B}_p} : r_s(\epsilon, t_{prob}^*) = -0.61$, both with $p \ll 0.05$. This correlation is a key insight that drives the further development of our framework.

This trend is further illustrated in Figures 5a and 5b, where the performance peaks at lower $t$ values ($t \leq 200$) become more pronounced as $\epsilon$ increases, particularly when $\epsilon \geq 9/255$. In contrast, for lower $\epsilon$ values, it is more beneficial to maintain accuracy by selecting a later painting step, such as $t = 1200$, rather than an earlier step like $t = 300$. Therefore, depending on the attack magnitude, it is necessary to stop the painting process at different steps to optimize the PCL performance.

This phenomenon is visualized in Figure 6 (as well as in Figures 9, 10, and 11 in the Appendix), illustrating the confidence levels of $PCL_{\mathcal{B}_p}$ for each class versus various $\epsilon$ values at different $t$

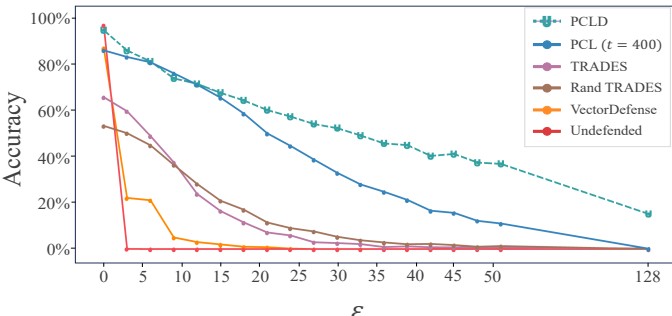

Figure 7: Test accuracy results against targeted-PGD$_{10}$ with increasing $\epsilon$ budget. A comparison between $PCL_{\mathcal{B}_p} D_{V^{\Leftarrow}}^{FC}$ and $PCL_{\mathcal{B}_p}$ with TRADES, Rand TRADES, VectorDefense and Undefended$_{\mathcal{B}_p}$ classifier. PCLD exhibits superior robustness, particularly at higher perturbations.

steps. A distinct trend is observed where, as $\epsilon$ increases: the value of $t$ at which the highest class probability shifts from the correct class to the targeted class occurs earlier. However, creating a rule-based decision system to stop painting at $t^*$ would involve numerous rules and careful handling of various edge cases. Therefore, training a decisioner model to learn these patterns is necessary.

## 4.4 DECISIONER CONTRIBUTION

We compare two decisioner architectures: a 1D convolutional network (Conv) and a fully connected network (FC). The Conv model uses two 1D convolutional layers with 64 filters (kernel size 3), followed by batch normalization, dropout, and adaptive max pooling. The FC model includes three fully connected layers of sizes 128, 64, and an output layer, each with ReLU activation and dropout. All models trained with cross-entropy loss and the SGD optimizer with a learning rate of 0.01, without employing a learning rate scheduler. We evaluate these models using three confidence matrices datasets derived from $PCL_{\mathcal{B}_p}$ on FGSM-attacked samples: targeted ($V^{\Rightarrow}$), untargeted ($V^{\Leftarrow}$), and combined ($V^{\Leftrightarrow}$), resulting in six total configurations. To address class imbalance, sample weights decrease as $\epsilon$ increases, adjusting the SGD loss per sample. Based on the validation results in Figure 12, we select the FC decisioner trained on untargeted FGSM inferences $V^{\Leftarrow}$.

The test performance of the selected decisioner is illustrated by the purple line marked with a "D" sign in Figure 5a. This ablation study highlights that the decisioner successfully learned the patterns and optimized the class decisions across paint steps ($t$). Figure 5b illustrates the performance of PCLD (dashed cyan line with the "U" sign) and PCL across all paint steps under a targeted-PGD$_{10}$ attack. Despite being trained only on one-step untargeted-FGSM, PCLD generalizes effectively to the iterative method, achieving the highest accuracy across nearly all perturbation magnitudes compared to all PCL paint steps. The advantage of incorporating the decisioner into PCL is further highlighted in Figure 7, where PCLD outperforms PCL$_{400}$ for most attack magnitudes, except at $\epsilon = 9$. A significant performance gap is observed at larger perturbations. Furthermore, in scenarios where maintaining high accuracy on benign images is critical, even the relatively small difference between PCLD and PCL$_{400}$ for $\epsilon = 0$ can be crucial.

## 4.5 COMPARING PCL AND PCLD WITH PRIOR ART

In this section, we evaluate the performance of the optimal PCL ($t = 400$) and PCLD configurations, consisting of the classifier trained on the benign images and their paints ($CL_{\mathcal{B}_p}$). For PCLD, we use the FC decisioner trained on untargeted FGSM attacks. The results are compared against state-of-the-art defenses, including TRADES, Rand TRADES, and VectorDefense.

Figure 7 shows the test performance of PCL and PCLD against targeted PGD$_{10}$ attack across various $\epsilon$ values. While preserving accuracy, PCLD consistently outperforms all other models, maintaining higher robustness overall attack magnitudes except for $\epsilon = 9$, where PCL is slightly better. Although TRADES and Rand TRADES were trained specifically on PGD$_{10}$, they struggle to maintain

Table 1: Benchmarking test accuracies (%) against various attacks under $\ell_\infty$. For each model we report accuracy under different adversarial attacks ($PGD_{10}$, $PGD_{100}$, $C\&W_{10}$, and AutoAttack) and inference time. PCLD achieves a superior balance between accuracy and robustness, demonstrating superior robustness at larger $\epsilon$ values.

| Model | Benign | $\epsilon = 8/255$ | | | | | $\epsilon = 20/255$ | Inference Time (Sec) |
| | | Gaussian Noise | $PGD_{10}$ | $PGD_{100}$ | $C\&W_{10}$ | AutoAttack | AutoAttack | |
|---|---|---|---|---|---|---|---|---|
| Undefended | 96.2 | 95.8 | 0.0 | 0.0 | 3.0 | 3.0 | 3.0 | 0.0003 |
| PCLD | **94.1** | **94.0** | 77.1 | **75.4** | 78.6 | 52.2 | **33.4** | 0.67 |
| PCL$_{400}$ | 85.5 | 85.5 | **77.7** | **76.6** | **80.5** | **58.3** | 21.9 | 0.1876 |
| TRADES (Zhang et al., 2019) | 65.4 | 65.2 | 37.3 | 36.7 | 35.1 | 37.5 | 20.9 | **0.0003** |
| Rand TRADES (Jin et al., 2023) | 53.0 | 52.9 | 37.7 | 37.3 | 32.8 | 31.0 | 18.2 | **0.0003** |

both accuracy and robustness, particularly against large perturbations, whereas PCLD demonstrates significantly greater resilience overall.

We attack VectorDefense adaptively, as described in Section 4.2, using a single surrogate model since VectorDefense outputs only the final canvas state for a given input. However, as provided in Figure 7, VectorDefense shows vulnerability, likely due to its reliance on only the final sketching step, omitting crucial intermediate stages. Enhancing VectorDefense with a reinforcement learning agent to strategically choose sketching strokes, could improve its robustness, allowing stopping the sketching at earlier optimal stages and even learning from the progression.

Table 1 shows that PCLD achieves a superior balance between accuracy and robustness compared to benchmarks. It maintains the highest benign accuracy (94.1%) and demonstrates strong resilience, particularly for large perturbations ($\epsilon = 20/255$), with 33.4% accuracy against AutoAttack, outperforming TRADES (20.9%) and Rand TRADES (18.2%). This suggests that PCLD's use of intermediate painting stages helps it manage the accuracy-robustness trade-off more effectively. While PCL$_{400}$ performs well in some cases, such as $PGD_{10}$ and $C\&W_{10}$, PCLD shows greater overall robustness, especially under higher $\epsilon$ values. Although TRADES and Rand TRADES offer faster inference times, their lower robustness highlights PCLD's superior performance in realistic settings.

In addition to the results shown in Table 1, for $\epsilon = 4/255$, PCLD achieves 68% accuracy under AutoAttack, further demonstrating its robustness at lower perturbation levels. Regarding computational complexity, the Painter requires 0.661 seconds, the Classifier takes 0.003 seconds, and the Decisioner operates in 0.001 seconds, resulting in a complete PCLD inference time of 0.67 seconds.

## 5 CONCLUSIONS

In this paper, we introduce the Painter-Classifier-Decisioner (PCLD) framework, a novel approach designed to reinforce adversarial robustness by leveraging stroke-based painting and adversarial training. Our empirical results on a subset of the ImageNet dataset show that PCLD achieves superior performance in both accuracy and robustness compared to state-of-the-art adversarial training against adaptive white-box attacks under the $\ell_\infty$ norm. PCLD addresses the generalization challenge. Initially trained on FGSM, PCLD demonstrates enhanced performance against PGD (Madry et al., 2018), C&W (Carlini, 2017) and AutoAttack (Croce & Hein, 2020). Moreover, PCLD maintains relatively high accuracy even as the attack strength increases, demonstrating its robustness across different perturbation levels. PCLD's ability to perform effectively in complex, high-resolution settings makes it a strong candidate for robust model deployment in real-world applications. However, a notable limitation of our approach is the computational complexity of employing an iterative painting model.

Future research can focus on optimizing the painter's efficiency and extending the evaluation to additional datasets, such as the entire ImageNet dataset, to further validate the framework's applicability. These advancements would pave the way for more resilient models capable of withstand sophisticated threats.

**Reproducibility statement:** The following GitHub repository includes the PCLD models, code, and links for the data: `https://github.com/pcld-defense/PCLD`

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

# A APPENDIX

## A.1 PAINTING PROCESS EXAMPLE

Figure 8: Painting process of all classes. The left column includes input photos, followed rightward by their respective paintings with $t$ strokes.

## A.2   DECISIONER MOTIVATION - PCL INFERENCE PATTERNS

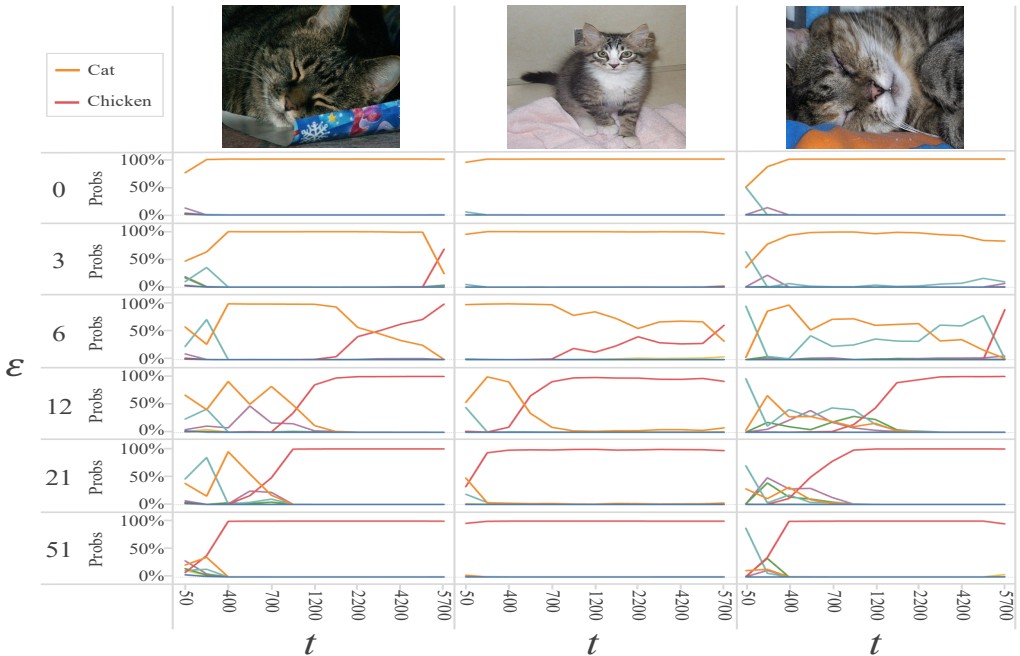

Figure 9: Softmax probability patterns of $PCL_{\mathcal{B}p}$ vs. targeted $PGD_{10}$ directed Cat $\Rightarrow$ Chicken for different epsilon budgets as a function of paint-step ($t$). As the perturbation increases, the shift in the highest class probability from the correct class to the targeted class occurs earlier, illustrating the dynamics of confidence variation with increasing $\epsilon$.

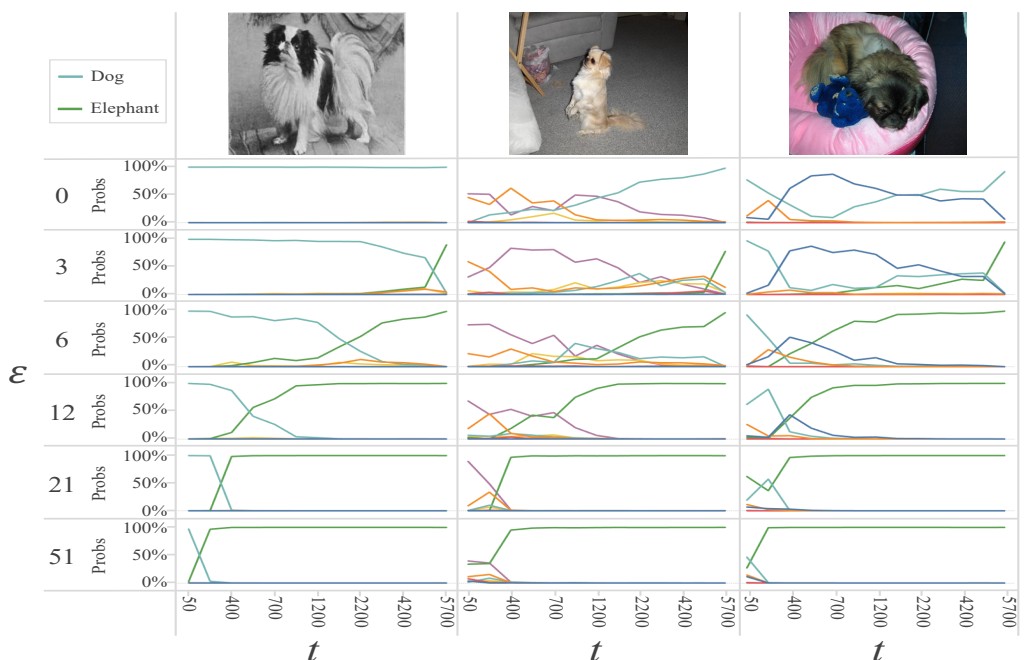

Figure 10: Softmax probability patterns of $PCL_{\mathcal{B}p}$ vs. targeted PGD$_{10}$ directed Dog $\Rightarrow$ Elephant for different epsilon budgets as a function of paint-step ($t$). As the perturbation increases, the shift in the highest class probability from the correct class to the targeted class occurs earlier, illustrating the dynamics of confidence variation with increasing $\epsilon$.

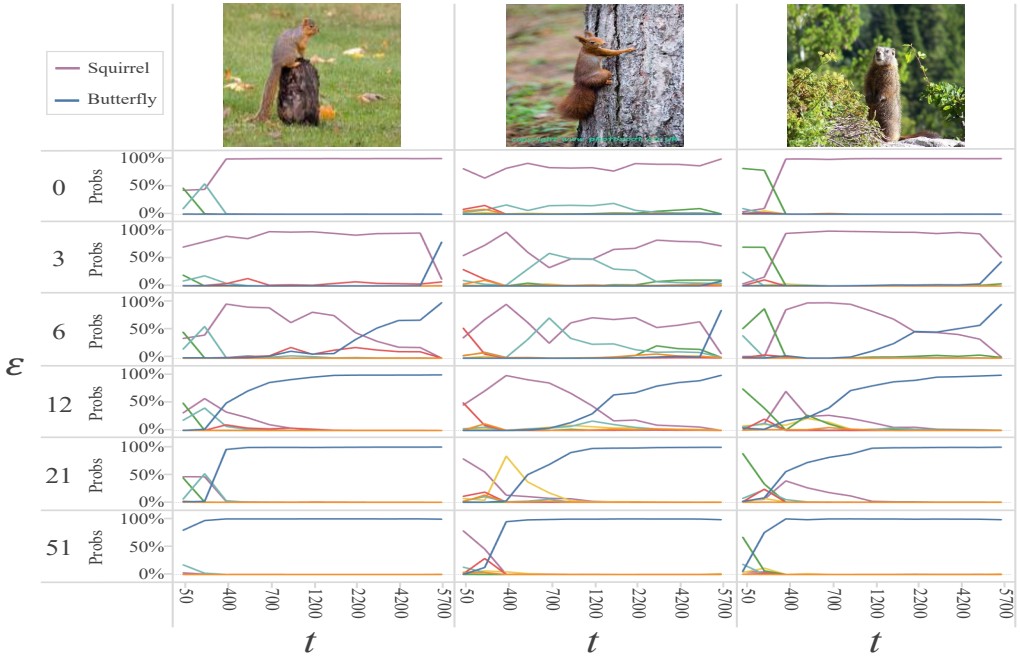

Figure 11: Softmax probability patterns of $PCL_{\mathcal{B}p}$ vs. targeted PGD$_{10}$ directed Squirrel $\Rightarrow$ Butterfly for different epsilon budgets as a function of paint-step ($t$). As the perturbation increases, the shift in the highest class probability from the correct class to the targeted class occurs earlier, illustrating the dynamics of confidence variation with increasing $\epsilon$.

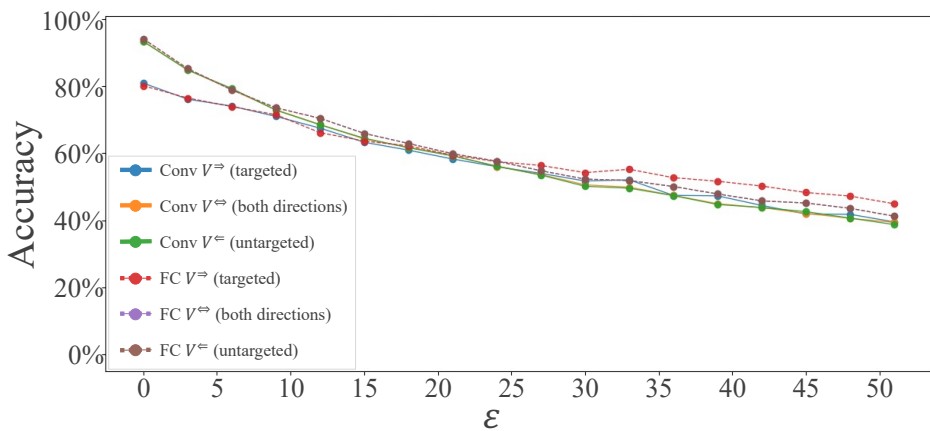

Figure 12: Comparison of different decisioner configurations. Specifically, two Decisioner architectures are compared, a 1D convolutional network "Conv" (markd with solid lines) and a fully connected network "FC" (markd with dashed lines), evaluated on three confidence matrices, retrieved from attacking $PCL_{\mathcal{B}_p}$ using: targeted ($V^{\Rightarrow}$), untargeted ($V^{\Leftarrow}$), and combined ($V^{\Leftrightarrow}$) FGSM-attack samples. The legend differentiates between the architectures and attack types. The FC model trained on untargeted FGSM ($V^{\Leftarrow}$) achieves the highest performance for $\epsilon \leq 24$.

