# OpenReview forum: "WHAT YOU PAINT IS WHAT YOU GET"
_ICLR.cc/2025/Conference — ICLR 2025 Conference Withdrawn Submission_

### Official Review · Reviewer_CqjU · 2024-10-29

**Soundness:** 2
**Presentation:** 3
**Contribution:** 2
**Rating:** 3
**Confidence:** 3

**Summary:**

In this paper, a defense approach against adversarial examples based on adversarial training and image painting techniques is proposed. More specifically, at test time, the input images are first reconstructed by a painter using different stroke levels; then, a subset of the reconstructed images are fed into the classifier, and finally, its output scores are provided to a neural network that performs the decision.

**Strengths:**

- The proposed approach is very interesting and clearly presented.
- The addressed topic is relevant, as adversarial examples still represent an open challenge.
- The authors designed an adaptive attack to evaluate their defense, applying suggestions from related works, such as the use of BPDA to approximate the gradients of components that hinder the optimization process. Although the experimental evaluation still presents some issues, I appreciate this effort.

**Weaknesses:**

- The evaluation of the defense needs many improvements. First of all, the need for BDPA to attack the defense should be better experimentally proven. Using only 10 iterations for PGD and CW attacks makes them too weak (they are usually run with at least 100 iterations, but they might be increased even to 1000). An end-to-end attack should be performed by using the strongest attack (e.g., AutoAttack), applying random restarts and EOT steps to see if actually the defense cannot be attacked in this setting.
The authors initially mentioned applying EOT, but this doesn't seem to be the case after reading the rest of the paper and the provided source code. EOT should be used when the defense method induces randomization. The authors do not mention the choice and/or tuning of the attack hyperparameter (e.g., step size, number of random restarts, if performed). Finally, I noted that the experiments report some strange results: on the undefended model, AutoAttack performs worse than the other weaker attacks, and the same happens for trades. How can it be explained? Could it be due to the non-original implementation used by the authors?
- The competing methods considered for comparison do not report state-of-the-art results. Please consider better-performing models (for instance, from RobustBench) and compare the defense with them.
- Training/validation/test splits are built from the ImageNet training data (I assume that based on the number of images, as the ImageNet validation set contains only 50 images per class, while here, each of the seven considered classes has 1000 samples). However, the defended models are pre-trained on the entire ImageNet training set. In this way, images from training data are used to test the model. This approach is not methodologically correct and poses concerns about the reliability of the provided results.
- The best-performing classifier is chosen based on its robust accuracy under attack. This approach is also not methodologically correct, as it is similar to performing model tuning on test data.
- The computational complexity of the proposed approach seems huge and poses serious concerns about its practical applicability in real-world settings. The authors only report the inference time without specifying how it is measured, which is over 600x and 2200x with respect to undefended and standard adversarially-trained models for PCL_400 and PCLD, respectively. As it seems to be the main limitation of the defense, I would expect more discussion and evaluations about it (see questions).
- The authors categorize defenses against adversarial examples into adversarial training and input transformation methods. However, several works (even complementary to them) are based on detecting adversarial inputs. For the sake of completeness, this category of defenses should at least briefly be mentioned.

**Questions:**

- How is the inference time computed?
- How much does each defense's component impact on it?
- What is the memory footprint of the defense when it is deployed (also compared to standard models)?

---

> ### Author Response · Authors · 2024-11-23
>
> Thank you for your detailed review and for pointing out areas for improvement. Please find our responses to your concerns below:
> **Weaknesses**
> - **Evaluation**\
> -- **BPDA Justification**\
> We would like to refer the reviewer’s attention to Figure 4 (left) which presents the experimental evidence for the need for BDPA to attack the defense. Furthermore, in preliminary experiments, we executed black-box attacks which did not succeed as well as with BPDA (was not reported due to limited relevance and length constraints).
> From a theoretical perspective, BPDA is a must due to obfuscated gradients of the painter (5200 iterations of actor & renderer models).\
> -- **Using or not using EOT**\
> We use EOT in all attacks except Naive-FGSM and Naive-PGD. We would like to refer the reviewer’s attention to line 359 in the paper and PCLD/model/pcld_bpda.py line 88 in the source code for applying EOT. Please note that our defense does not induce randomness. Our reason for using EOT is not a standard one. Instead of averaging gradients over random trials, we average gradients over the painting steps, i.e., $g ∈ R^{T×W×H×C}$ are averaged over the painting steps dimension $T$ to match the input shape $x ∈ R^{W×H×C}$.\
> -- **Number of attack iterations & using AA**\
> We conducted evaluations with PGD-100 using BPDA + EOT; see Table 1 on page 10. Carlini et al. [1] suggested that the number of iterations should be increased (to 100, 1000, etc) until additional iterations improve the attack. We see that the improvement of the attack from 10 to 100 is small. Therefore we believe that increasing the number of iterations by additional order of magnitude is not required.\
> We agree with the reviewer that there is a need for the strongest attack (e.g., AutoAttack), applying random restarts and EOT. Therefore, we utilized AutoAttack (AA), which includes two versions of APGD with 100 iterations and 5 random restarts, DeepFooll with 100 iterations, and Square with 5000 iterations and 5 random restarts.\
> -- **Attack Hyperparameters and Clarification**\
> The step size for PGD was set to ϵ/number of iterations. For all other attacks, we used the default values for step sizes, random restarts and confidence (in CW), which are available in our code. We will add this information to the appendix for completeness.\
> -- **AA vs. CW and PGD**\
> We double-checked our results and confirmed their accuracy. The Adversarial Robustness Toolbox (ART) library we used claims to implement the CW and AutoAttack (AA) methods faithfully, as per their original publications. To our knowledge, there are no existing references that evaluate both AA, CW/PGD on the same defense for the ImageNet dataset under the $l_\inf$. Therefore, no benchmark exists for direct comparison in this scenario.\
> -- **Competing methods**\
> We selected VectorDefense as it is the only other stroke-based defense, making it a suitable comparison for our painting approach. TRADES and its improved variant were chosen due to their focus on the accuracy-robustness tradeoff, which aligns with our defense's goal. We believe these baselines provide meaningful insights but acknowledge the value of comparing to models in RobustBench. \
> The [latest leaderboard performance on ImageNet](https://robustbench.github.io/#div_imagenet_Linf_heading) shows:
> - **81.48%** best benign accuracy and **59.64%** best robust accuracy with $l_\inf=4/255$.\
> In our paper (see **line 486 - Table 1**), we report the following results:
> - PCLD: **94.1%** benign accuracy and **52.2%** robust accuracy with $l_\inf=8/255$.
> - PCL:  **85.5%** benign accuracy and **58.3%** robust accuracy with $l_\inf=8/255$.\
> Note the larger attack budget and preferable benign accuracy.
> We conducted additional evaluations with $l_\inf=4/255$. We will report $4/255$ results shortly.\
> -- **Train/Val/Test**\
> We understand your concern about the training, validation, and test splits. To clarify, we constructed our dataset by selecting sub-classes from the ImageNet training set and grouped them under broader categories. For example, sub-classes like "cairn terrier" "French bulldog" and "Tibetan mastiff" were relabeled as "dog" effectively creating new classes distinct from the original ImageNet labels.
> The pretrained ResNet-18 was then fine-tuned on these new classes, with distinct images used for training, validation, and testing to prevent data leakage. Additionally, the decisioner was trained entirely from scratch, learning from the newly defined categories and resized images, further mitigating potential biases.
> We consistently applied this approach across all benchmarks. We hope this alleviates your concerns about methodological correctness and data leakage.\
>
> [1] Carlini, Nicholas, et al. "On evaluating adversarial robustness." arXiv preprint arXiv:1902.06705 (2019).

---

> ### Author Response · Authors · 2024-11-23
>
> -- **Choosing best performing clf**\
> We disagree with the methodological concern related to comparison to the best-performing classifier based on its robust accuracy. Please note that the best-performing classifier is used here to showcase a favorable classification in the absence of a decisioner. Said that, all hyperparameter tuning and model selection were performed exclusively on the validation set, not on the test set. In the paper, we presented test results that are nearly identical to those on the validation set, due to space limitations. The validation set was used for all model optimizations, and no part of the test data was used in these processes.\
> -- **Computational complexity**\
> We measured inference time by averaging the time taken for each image from input to output. Specifically, the times are: Painter = 0.65 sec, Classifier = 0.003 sec, Decisioner = 0.0005 sec, and complete PCLD = 0.67 sec. These and further runtime analysis results will be included in the appendix.
> The longer runtime is due to the Painter component. This complexity affects both attackers and defenders, not just defenders. While this approach might not be suited for all real-time applications, a response time around 0.5-0.67 seconds can still be practical for various use cases. This work aims to demonstrate the potential of painting as a defense, paving the way for further research to optimize its efficiency.\
> -- **Additional Defense Category**\
> Thank you for pointing this out. We will briefly include detection-based defenses in the discussion, as they complement adversarial training and input transformation methods by identifying adversarial inputs, adding an important layer of security.\
> -- **Memory Footprint**\
> Our method does have increased memory usage due to intermediate canvases generated by the Painter, compared to standard adversarially-trained models. However, this added cost brings substantial gains in robustness. Future work may focus on optimizing the number of canvases required, reducing memory while maintaining security.

---

> ### Author Response · Authors · 2024-11-24
> **RE: RobustBench**
>
> AA with $L_∞$ = 4/255:
> Accuracy = 68%

---

> > ### Author Response · Authors · 2024-11-26
> > **Relevant Changes to the Paper**
> >
> > We have addressed your comments and made the following changes in the paper:\
> > 1. We reported the accuracy of PCLD under AutoAttack with $\epsilon=4/255$, achieving 68% accuracy.\
> > Location: Section 4.5, end of results.\
> > 2. We detailed the hyperparameters for the attacks.\
> > Location: Section 4.1, line 259.\
> > 3. We reported the computational complexity for each component.\
> > Location: Section 4.5, end of results.\
> > 4. We briefly discussed detection-based defenses.\
> > Location: Section 1, line 45.

---

> > > ### Comment · Reviewer_CqjU · 2024-11-26
> > >
> > > I thank the authors for their detailed answers, which shed light on several aspects. I have provided some comments and requests for further clarification below.
> > >
> > > ### **Attack Hyperparameters and Clarification - Computational complexity - Additional Defense Category - Memory Footprint**
> > > Thanks for taking into account my comments and for the clarifications.
> > >
> > > ### **Choosing best performing clf**
> > > I see the point from the authors and acknowledge that the followed methodology is correct.
> > >
> > > ### **Using or not using EOT**
> > > I thank the authors for clarifying this aspect. In the paper, EOT is mentioned only once in Sect. 4.2 - so I guessed from that that it was applied as usual (averaging the losses over input transformations to overcome randomization introduced by the defense). I suggest the authors add in the 2nd paragraph of Sect. 4.3.1 a reference to the EOT method to make clear to the readers the link between EOT and the way it is implemented.
> > >
> > > ### **AA vs. CW and PGD**
> > > I refer the authors to the AutoAttack paper, where the different attacks were compared, and another recent work [a]. Indeed, AutoAttack leverages APGD, which is an improved version of PGD: for this reason, I found it weird that PGD performed better than it in some cases. Overall, AutoAttack's superiority is usually achieved by its ability to overcome some optimization issues that affect standard stochastic gradient descent methods used by PGD and CW attacks.
> > > Note that in its original implementation, AutoAttack executes the following attacks: untargeted APGD with CE loss, targeted APGD, targeted FAB and Square. The implementation provided in the Art library runs the following ones: untargeted APGD with CE loss, untargeted APGD with DLR loss, DeepFool and Square. It is thus likely that using the original implementation of AutoAttack might produce different results.
> > >
> > > [a] Cinà, A.E., Rony, J., Pintor, M., Demetrio, L., Demontis, A., Biggio, B., Ayed, I.B., & Roli, F. (2024). AttackBench: Evaluating Gradient-based Attacks for Adversarial Examples. ArXiv, abs/2404.19460.
> > >
> > > ### **Train/Val/Test**
> > > Thanks for the clarification. However, my concerns partially remain: the problem here is that the pretrained model already "saw" during their training the images used to test the defense. It would be better to use disjoint datasets.
> > >
> > > ### **Competing methods**
> > > RobustBench models are trained on the full 1000-classes ImageNet training set and tested with the ImageNet validation set. The defense in this paper is evaluated under a completely different setting (which, moreover, is an easier classification task), and thus it is not possible to directly compare the results, even to provide rough estimates. The comparison must be performed using the same experimental setting (not only the perturbation size).
> > >
> > > ### **Number of attack iterations & using AA**
> > > I disagree with the author, as using only 10 iterations for gradient-based attacks will likely not lead the optimization to converge. Indeed, in the referenced paper [1] the authors point out that, even if some attacks are successful with a few iterations, this is not common and usually the lower iteration number is set to 100 - and might grow even to over 1000. It is unlikely that by increasing the number of iterations from 10 to a higher order of magnitude, the attack success rate will not improve - at least some optimization issue is present. Given that the use of BPDA + EOT addressed - based on the authors' findings - shattered gradients caused by the painter, the results in Tab. 1 may reveal other pitfalls in the optimization process (e.g. a not properly tuned step size). Some checks that could be helpful to provide further insights should try to answer the following questions: after how many iterations does the optimization get stuck (i.e., the attack loss does not improve anymore)? What happens when the attack step size is changed? Is the attack able to push images towards the maximum allowed Linf ball? What is the difference between using targeted and untargeted PGD?
> > >
> > > ### **BPDA Justification**
> > > The abovementioned point was one of the reasons that made me ask for more support for the need to use BPDA. Of course, in the presence of gradient obfuscation, the optimization might get stuck after a few iterations. Fig. 4 shows interesting results but is only a partial view of the problem (see the last part of the previous point for what could be done to provide further insights on that). Note that I'm not questioning the need for BPDA - conversely, I appreciate the effort made by the authors to assess a reliable robustness evaluation - but I'm asking for more empirical justification.

---

> > > > ### Author Response · Authors · 2024-12-02
> > > >
> > > > Thank you for your thoughtful review. We appreciate your insights, which help us improve our work. We are excited to present our research at the conference, aas we believe that further discussions with peers will also advance the development of this idea. Below, we address your concerns and provide clarifications to support the potential of our contribution.\
> > > >
> > > >
> > > > **Using or not using EOT**\
> > > > Thanks for this suggestion. We have added a reference to the EOT method in the second paragraph of Section 4.3.1, which links it more clearly to our specific implementation.\
> > > > **AA vs. CW and PGD**\
> > > > Thank you for pointing to the Cinà et al. (2024) paper (AttackBench). We reviewed it and have updated the revised version of our paper (the last sentence in Section 4.5) to mention that, according to their findings, the ART implementation of APGD may produce suboptimal results. Specifically, APGD in ART ranks lower compared to other implementations, which likely explains why PGD outperforms AutoAttack in some cases in our experiments.\
> > > > **Train/Val/Test**\
> > > > Thank you for your feedback. To clarify, the classifier was trained, validated, and tested mostly on painted versions of the data, which helps prevent data leakage. Likewise, the decisioner was trained, validated, and tested using confidences derived from these painted versions.\
> > > > **Number of attack iterations & using AA**\
> > > > Thank you for your feedback. We agree that increasing iterations can improve attack success rates, and the benchmarks in the paper show similar trends. For example, TRADES (Zhang et al., 2019), accuracy drops from 37.3% with PGD-10 to 36.7% with PGD-100, and Rand TRADES (Jin et al., 2023) from 37.7% to 37.3%. While further exploration of higher iterations and step size tuning is valuable, we believe it wouldn't significantly alter our conclusions. Given the high computational cost, we plan to explore this in future work.\

---

> > > > > ### Author Response · Authors · 2024-12-02
> > > > >
> > > > > We have made the suggested updates to the paper, but unfortunately, the revision period has ended, and we cannot upload the updated version.

---

### Official Review · Reviewer_TYXM · 2024-10-31

**Soundness:** 2
**Presentation:** 2
**Contribution:** 2
**Rating:** 3
**Confidence:** 4

**Summary:**

This paper introduces PCLD, a novel defensive framework designed to mitigate adversarial attacks.
PCLD leverages the painter algorithm to generate images at varying levels of detail based on the number of strokes applied. The main assumption is that the painting algorithm would start building the image from the non-adversarial content, thus producing a "clean" image. These "painted" representations are then integrated with adversarial training to robustly train both a classifier and a decisioner, based on the classifier scores.

**Strengths:**

- usage of painting algorithm as a defensive transformation

**Weaknesses:**

- Core theoretical concepts missing
- Experimental evaluation should be enhanced
- Missing implementation details

**Clarity.** The authors omitted the definition of core concepts, that are essential for the reader, such as adversarial attacks and adversarial training.
Their framework comprises adversarial training, however, the details behind the choice of the training loss are not discussed.
There is no discussion about the optimizer and/or scheduler used, nor about their hyperparameters (except when they describe how they applied BPDA, using the encoder-decoders).
The state-of-the-art does not include methods that employ generative models that make use of style transfer to images to improve adversarial robustness [1].
Additionally, the authors should clearly discuss why the evaluation is limited at L-infinity attacks, and if this is a limitation of the defense.

**Evaluation should be improved.** All the accuracy-perturbation curves are made using PGD-10, while these results can be discarded, according to [2]. The results of PGD-100 are presented, but it is not clear if the adaptive strategy has been used or not. Additionally, as reported, having the decisioner (PCLD) does not really help improving the robust accuracy.
More details are given in the questions, and these are the main points to be addressed to improve the score of this review.

[1] Lin, Hubert, et al. "What can style transfer and paintings do for model robustness?." Proceedings of the IEEE/CVF Conference on Computer Vision and Pattern Recognition. 2021.
[2] Carlini, Nicholas, et al. "On evaluating adversarial robustness." arXiv preprint arXiv:1902.06705 (2019).

**Questions:**

- Would it be possible for an attacker to create an image for which painting the initial strokes produces the adversarial perturbation? If the painting is ML-based, probably there is a chance of having a successful adaptive attack in this way.
- What happens when an attacker targets the decisioner?
- In the abstract, what are "provable adaptive attack techniques?"
- The authors did not compare their approach with data augmentation involving another type of filtering, e.g., low-pass filter, to exclude high-frequency components of the images?
- How is this approach different than, e.g., JPEG compression, which was proven ineffective as a defense?
- The authors should provide the result of the attack with an infinite budget, to show that the adaptive attack reaches 100% ASR as per the cited guidelines [2]

---

> ### Author Response · Authors · 2024-11-16
>
> Thank you for your detailed review and for highlighting both the strengths and areas for improvement in our work. We address each of your concerns and questions below:\
> **Weaknesses- Clarity**
> - In **Section 4.4** we report on the six Decisioner configurations.\
> We use **cross-entropy loss** for training the Decisioner adversarially, with **SGD** optimizer (lr=0.01) without **scheduler**. These details will be included in the revision. The git repo contains this config ready to run.
> - Due to the limited space and substantial results we had to choose the experiments to report. We have chosen $L_∞$ because it is more favourable with the attacker than $L_2$ and it is the standard in [RobustBench Leaderboard for ImageNet](https://robustbench.github.io/#div_imagenet_Linf_heading).\
> Since $||x||_2$ is always at least as high as $||x||_∞$ and can be as high as $\sqrt n ||x||_∞$, an attack vector that would be allowable under $||x||_∞$ may be too large under $||x||_2$ resulting in lower perturbations overall.\
> To show it empirically and address your concerns, we run (in the last couple of ours) a PGD100 attack under $L_2$ with a similar budget to $L_∞: 8/255$, that is $\epsilon=16$, here is the math:\
> $||x||_2 = \sqrt n ||x||_∞$ $\Longrightarrow$ $\epsilon_2 = \sqrt n \cdot \epsilon_∞$\
> In our case: $\sqrt n = \sqrt (300x300x3) = \sqrt 27000 \\approx 520$\
> Finally:\
> $\epsilon_2 = 520 \cdot 8/255 = 16$\
> The results we got for $\epsilon_2 = 16$ are almost identical to $\epsilon_∞ = 8/255$: \
> **76.1%** accuracy (for $\epsilon_∞ = 8/255$ we reported **75.4%** accuracy).\
>
> **Weaknesses- Evaluation**
> - We conduct evaluations using PGD-100. As discussed in [1], doubling the number of iterations to verify if the chosen number is sufficient. In our case, moving from PGD-10 to PGD-100 showed only minor changes.\
> All evaluations in the paper used the adaptive attack described in Section 4.2, except for 4.3.2 (following the sanity checks proposed in [1]).\
> Note that AA combines 100 and even 5000 iterations.
> - The use of the Decisioner improves robustness in larger attack budgets (See Table 1 - AA with $\epsilon=20/255$ and also Figure 7 - $\epsilon>=12$). It also improves benign accuracy (As reported in Table 1), “manage the accuracy-robustness trade-off more effectively” (line 513).\
>
> **Q1**\
> Targets the policy of the RL painter would require significant advances in attack methodologies. Designing such an attack is beyond the current scope of our work. However, we see potential value in this direction and would be happy to discuss it further.\
> **Q2**\
> If the question concerns attacks targeting all components of PCLD, including the Decisioner, we addressed this in Section 4.4 (Figure 7) and Section 4.5 (Table 1), where adaptive attacks were conducted on the entire system, , following the recommendations in [1, 2]. However, if the question refers to attacking only the Decisioner, this would be impractical since it relies on the Classifier's output, which means generating a denoised probability vector. Note that in Section 4.3.2, we tackled a naive baseline where attacks were generated using gradients from the CLassifier-Decisioner (CLD) component.\
> **Q3**\
> As discribed in Section 4.2, we developed adaptive attack strategy based on BPDA+EOT [2]. We further conduct sanity checks in Section 4.3.2 to our method as advised in [1] and prove its ability to produce useful gradients from the painter.\
> **Q4**\
> We focused on novel painting-based transformations, similar to recent works like VectorDefense that use sketching against adversarial attacks. While comparing with filtering techniques or other augmentation methods like randomized smoothing could be valuable, our primary aim was to explore this new approach. Future research could extend these comparisons.\
> **Q5**\
> JPEG compression and similar early filtering defenses were broken using adaptive attacks like BPDA + EOT [2]. In our study, we applied an adaptive attack based on BPDA + EOT to PCLD, and our results show that PCLD remains resilient, unlike earlier filtering defenses that proved ineffective against these sophisticated attacks.\
> **Q6**\
> According to [1], “it is not possible to do better than random guessing with a L∞ distortion of 0.5: any image can be converted into a solid gray picture”. In our paper, we evaluated PCLD using L∞​ distortions with ϵ=128/255 and showed that accuracy dropped to random guessing levels, as expected. If desired, we could extend this evaluation to even higher distortions (e.g., ϵ=510/255).\
>
> [1] Carlini, Nicholas, et al. "On evaluating adversarial robustness." arXiv preprint arXiv:1902.06705 (2019).\
> [2] Anish Athalye, Nicholas Carlini, and David Wagner. Obfuscated gradients give a false sense of security: Circumventing defenses to adversarial examples. In International conference on machine learning, pp. 274–283. PMLR, 2018.

---

> ### Author Response · Authors · 2024-11-24
>
> Dear Reviewer TYXM, As we near the end of the discussion phase, we would like to query whether we have sufficiently alleviated your concerns. If anything remains unclear, we would be happy to provide further clarification.

---

> > ### Author Response · Authors · 2024-11-26
> > **Relevant Changes to the Paper**
> >
> > We have addressed your comments and made the following changes in the paper:\
> > We provided the details of our training setup for the CLassifiers and Decisioners.\
> > Locations:\
> > Section 4.3.1, line 356 (Classifiers).
> > Section 4.4, line 427 (Decisioners).

---

> > > ### Comment · Reviewer_TYXM · 2024-11-28
> > > **Response to Authors**
> > >
> > > I thank the authors for their clarification and additional experiments.
> > >
> > > * My concern still stands on the adaptive attack (attack aware of the defense in place and targets its weakest point), which is of key importance when proposing defenses. The idea is that the model should be affected by this attack, but in a limited manner w.r.t. an undefended model. This proves the decision boundary is still better than no defense at all. This is a key aspect and it's not addressed yet.
> > > * Evaluation concerns are also part of other reviewers' comments, and I agree with these.
> > >   - Number of steps
> > >   - potential gradient obfuscation issues
> > >   - potential bugs in the attack implementations
> > >   - extensibility to other perturbation models
> > >
> > >
> > > While I thank the authors for the effort put in the rebuttal, my concerns are still not fully addressed. I hope the authors can improve this work further with my suggestions and the feedback received by me and the other reviewers. I will keep my score.

---

### Official Review · Reviewer_Lyyx · 2024-11-03

**Soundness:** 3
**Presentation:** 3
**Contribution:** 3
**Rating:** 6
**Confidence:** 4

**Summary:**

This paper proposed Painter-CLassifier-Decisioner (PCLD), an adversarial defense framework that introduces stroke-based painting to adversarial training. PCLD obtains a good generalization, able to defend against various adversarial attacks with different attack budgets. Compared with the baseline methods, PCLD helps improve both the clean accuracy and robust accuracy.

**Strengths:**

- The paper is well-written and easy to understand.
- The design of the defense pipeline is intuitive and makes sense to me. I think leveraging stroke-based painting to remove adversarial noise is an interesting idea.
- As a plug-and-play method, it obtains better robustness than some baseline methods using adversarial training.

**Weaknesses:**

- The inference time of PCLD is significantly longer than baseline methods.
- Compared with many methods on RobustBench[1], there is still a non-negligible performance gap.

[1] https://robustbench.github.io/

**Questions:**

- In Fig. 2, does the decisioner learn the varying trend of the probability of each class and output a new probability or just choose one probability instance as the final output?
- Does the selection of the painting steps (including the total number of steps) affect the performance a lot? In this paper, based on what kind of metric do you set the painting steps?
- I am wondering if applying PCLD to a robust classifier (e.g. trained via TRADES) would further help improve the robustness.
- Is PCLD still effective against $l_2$-norm attacks? In this paper, it is only evaluated on $l_\infty$-norm attacks.
- For PCL without a decisioner, how do you get the final output? By manually setting a painting step and choosing this canvas as the input and get its output?
- According to Sec 4.3, $PCL_{B_p}$  stands for training on a set consisting of both clean and painting images. What is $PCL_{400}$ in Sec 4.4?

---

> ### Author Response · Authors · 2024-11-16
>
> Thank you for your constructive feedback.
>
> 1. **Weaknesses: PCLD inference time**\
> The higher running time is due to the Painter, which is integral to our method. A faster Painter would make PCLD more efficient. Notably, longer running times impact both defenders and attackers. A response time of ~0.5 seconds is reasonable for many interactive use cases, though unsuitable for autonomous driving or search engines. Our work provides motivation for developing a faster Painter. This is the first to demonstrate painting's potential in adversarial defense, If the reviewers see the promise in this approach, subsequent works could focus on developing faster painting schemes to build on this foundation.
>
> 2. **Weaknesses: Performance Gap on RobustBench**\
> We strongly disagree with the reviewer and kindly ask that this weakness be reevaluated.
> The [latest leaderboard performance on ImageNet](https://robustbench.github.io/#div_imagenet_Linf_heading) shows:
> - **81.48%** best benign accuracy and **59.64%** best robust accuracy with $l_\inf=4/255$.\
> In our paper (see **line 486 - Table 1**), we report the following results:
> - PCLD: **94.1%** benign accuracy and **52.2%** robust accuracy with $l_\inf=8/255$.
> - PCL:  **85.5%** benign accuracy and **58.3%** robust accuracy with $l_\inf=8/255$.\
> Note the larger attack budget and preferable benign accuracy. We will report $4/255$ results shortly.
>
> 3. **Q1: Decisioner Output**\
> The Decisioner is a classifier that, given the varying trend over all painting steps of the probabilities, outputs a new probability vector. The final decision is made by selecting the highest probability class.
>
> 4. **Q2: Selection of the painting steps**\
> We chose the painting steps to balance robustness and efficiency following the intuition provided in line 50. Early steps start at 50 and increase to 200 by increments of 50, capturing significant changes (as the initial strokes are large). Beyond 200, the increments expand to 500 up to 5200, focusing on more gradual changes. This approach is detailed in **Line 322, Section "4.3.1 TRAINING CLASSIFIER WITH PAINTS".**\
> It was impractical to experiment with many options. Since this is a novel approach, our method lays the groundwork for future exploration of different step configurations to assess their impact.\
> We hope this clarifies our choices.
>
> 5. **Q3: Combine PCLD with Robust Classifier (e.g., TRADES)**\
> PCLD can be easily combined with other robust training methods. The main contribution of TRADES is handling the **robustness-accuracy tradeoff**, already handled by PCLD. Furthermore, PCLD addresses other challenges that TRADES does not, such as the generalization issue. While trained only on FGSM, PCLD shows SOTA performance on stronger attacks.\
> In future work, we intend to explore alternative approaches for training the Decisioner.
>
> 6. **Q4: The effectiveness of PCLD against L2 attacks**\
> Due to the limited space and substantial number of results we had to carefully choose the experiments to report. When considering $L_∞$ vs $L_2$ we have chosen $L_∞$ because it is more favourable with the attacker than $L_2$.  Since $||x||_2$ is always at least as high as $||x||_∞$ and can be as high as $\sqrt n ||x||_∞$, an attack vector that would be allowable under $||x||_∞$ may be too large under $||x||_2$ resulting in lower perturbations overall. Furthermore, we choose $L_∞$ because it is a widely accepted benchmark -- see [RobustBench Leaderboard for ImageNet](https://robustbench.github.io/#div_imagenet_Linf_heading).\
> Bottom line, every perturbation allowable under $L_2$ norm with certain $\epsilon$ is allowable under $L_∞$ with a lower $\epsilon$. To show it empirically and address your concerns,  we run (in the last couple of ours) a PGD100 attack under $L_2$ with a similar budget to $L_∞: 8/255$, that is $\epsilon=16$, here is the math:\
> $||x||_2 = \sqrt n ||x||_∞$ $\Longrightarrow$ $\epsilon_2 = \sqrt n \cdot \epsilon_∞$\
> In our case: $\sqrt n = \sqrt (300x300x3) = \sqrt 27000 \\approx 520$\
> Finally:\
> $\epsilon_2 = 520 \cdot 8/255 = 16$\
> The results we got for $\epsilon_2 = 16$ are almost identical to $\epsilon_∞ = 8/255$: \
> **76.1%** accuracy (for $\epsilon_∞ = 8/255$ we reported **75.4%** accuracy).
>
> 7. **Q5: Painter-CLassifier (PCL) Output**\
> For PCL (without the decisioner), the final output is determined by manually selecting a specific painting step, which serves as the final canvas for classification. Based on our analysis of the validation results from attacking PCL, we chose $t=400$ as the optimal step. In the paper, we present the test results in **Figure 5**, as they are nearly identical to the validation results.\
> All hyperparameter optimizations were conducted exclusively on the validation set.
>
> 8. **Q6: Painter-CLassifier (PCL) Naming (PCL400 vs PCLBp)**\
> $PCL_400$ represents the best performing PCL variant $$PCL_{B_p} $$ with the optimal paint step we choose after the analysis provided in Section **4.3.3**, that is t=400.

---

> ### Author Response · Authors · 2024-11-20
> **AA with $L_∞$ = 4/255 Status**
>
> Accuracy = 68%.

---

> ### Author Response · Authors · 2024-11-24
>
> Dear Reviewer Lyyx, As we near the end of the discussion phase, we would like to query whether we have sufficiently alleviated your concerns. If anything remains unclear, we would be happy to provide further clarification.

---

> > ### Author Response · Authors · 2024-11-26
> > **Relevant Changes to the Paper**
> >
> > We have addressed your comments and made the following changes in the paper:\
> > We reported the accuracy of PCLD under AutoAttack with $\epsilon=4/255$, achieving 68% accuracy.\
> > Location: Section 4.5, end of results.

---

> ### Comment · Reviewer_Lyyx · 2024-11-26
> **Reply to the Authors**
>
> I greatly appreciate the responses from the authors, which have addressed my concerns. Therefore, I have raised my rating to 6.

---

### Official Review · Reviewer_kJbZ · 2024-11-05

**Soundness:** 2
**Presentation:** 2
**Contribution:** 2
**Rating:** 5
**Confidence:** 3

**Summary:**

The paper proposes a reinforcement learning based painting algorithm framework to defend against adversarial perturbations in images. The framework includes a step to recreate the perturbed image in gradual steps using painting strokes and a classification-decision system to make decisions based on intermediate steps of painting.

**Strengths:**

This approach is a novel application of painting algorithms for transformation-based defenses against adversarial attacks.

**Weaknesses:**

The comparison of running times (0.67 vs 0.0003) with prior work is a huge downside which makes this method quite impractical to be deployed extensively.

**Questions:**

1. (Line 309) Some use of acronyms like CL for PCL and CLD for PCLD are not clear.

2. It wasn't entirely clear to me which part is being referred to as adversarial training.

3. Why is a non standard image size like 300x300 used for the ImageNet experiments. This would probably make intermediate activations in the model quite different to the standard 224x224 size used for ImageNet in other work.

---

> ### Author Response · Authors · 2024-11-15
>
> Thank you for your thoughtful feedback. We appreciate your acknowledgment of the novelty of our approach. Below, we address each of your comments and concerns in detail.
>
> 1. **Comparison of Running Times**: The higher running time is largely due to the Painter component, which is integral to our method. If a faster Painter were developed, PCLD would also become more efficient. Importantly, longer running times affect both defenders and attackers equally, not only defenders. A response time of approximately 0.5 seconds is reasonable for many interactive use cases, though it may not be suitable for applications like autonomous driving or search engines. Until now, there may have been little incentive to develop a faster Painter, but our work provides a compelling motivation to do so.\
> This is the first work to demonstrate the potential of painting as a defense mechanism in adversarial settings. If the reviewers see the promise in this approach, subsequent works could focus on developing faster painting schemes to build on this foundation.
>
> 2. **Acronym Clarification (CL and CLD)**: We apologize for the confusion regarding the acronyms. As shown in Figure 2, "CL" represents the **Classifier**, "P" is the **Painter**, and "D" is the **Decisioner**.
> In Section "4.2 ADAPTIVE ATTACK STRATEGY" we explain how we evaluate our adaptive attack strategy using the prescriptions appear in **[1]**. Then in Section “4.3.2 ADAPTIVE ATTACK - SANITY CHECK” we show the results.
> In line 309, we refer to the following models:
> - **Partial Model (PCL)**: This model includes only the Painter and Classifier components.
> - **Full Model (PCLD)**: This model includes the Painter, Classifier, and Decisioner components.
> Due to the Painter component, the **gradients for these models are not automatically useful** for an adversary. Therefore, a naïve adversary that does not use an adaptive attack may produce ineffective perturbations by ignoring the Painter’s impact. In such cases:
> - For the **PCL** model, a naïve attack would use only the **Classifier (CL)** to generate perturbations.
> - For the **PCLD** model, a naïve attack would use the **Classifier-Decisioner (CLD)** components, excluding the Painter.
>
> We will clarify this in the manuscript to ensure this distinction is clearly communicated.
>
> 3. **Adversarial Training Component**: Our adversarial training approach differs from traditional methods. Rather than directly training on adversarially perturbed inputs, we adversarially train the Decisioner by focusing on the classifier’s responses at various stages of the painting process w.r.t benign inputs and FGSM attacked inputs at various epsilon budgets.
> The process:
> - Generating attacks (with the adaptive attack) using the partial model “Painter-CLassifier (PCL)” from the training data.
> - The inferences (confidences) matrix of the classifier for the attacked inputs at various painting steps is used as a training dataset for the Decisioner.
> This enables the Decisioner to recognize adversarial patterns by observing the classifier’s behavior over progressively denoised versions of the image.
> **See**:
> **Line 431** in Section “4.4 DECISIONER CONTRIBUTION”: “Based on the validation results in Figure 12, we select the FC decisioner trained on untargeted FGSM inferences V⇐”
> **Line 195** in Section “2.4 DECISIONER”: “A decisioner trained on adversarial examples of the confidence matrices V should learn to identify such patterns and select the right class.”
>
>
> 4. **Choice of 300x300 Image Size**: The choice of a 300x300 image size was arbitrary. It is a configurable parameter that does not significantly affect model performance. We experimented with this size for consistency, but the framework is flexible and can easily adapt to other dimensions (like 224x224) without compromising results.
>
> [1] Nicholas Carlini, Anish Athalye, Nicolas Papernot, Wieland Brendel, Jonas Rauber, Dimitris Tsipras, Ian Goodfellow, Aleksander Madry, and Alexey Kurakin. On evaluating adversarial robustness. arXiv preprint arXiv:1902.06705, 2019.
>
> ------
> ------
> We thank the reviewer for the valuable feedback which helps us to fine-tune the the presentation of the manuscript and hunt down any remaining unclarities.

---

> > ### Author Response · Authors · 2024-11-24
> >
> > Dear Reviewer kJbZ, As we near the end of the discussion phase, we would like to query whether we have sufficiently alleviated your concerns. If anything remains unclear, we would be happy to provide further clarification.

---

> > ### Author Response · Authors · 2024-11-26
> > **Relevant changes to the paper**
> >
> > We have addressed your comments and made the following changes to improve the clarity and completeness of the paper:\
> > **Clarification of P|CL|D Components:**\
> > We clarified the CL and CLD acronym.\
> > Location: Section 4.2, line 310.

---

> ### Comment · Reviewer_kJbZ · 2024-11-26
>
> Thank you for addressing my concerns, especially explaining the adversarial training in detail. I have an additional request which will help improve the presentation. I would appreciate it if all the figure captions (especially the ones currently in the appendix) were expanded and made self-contained so that the result summary is communicated without referring back to the text multiple times. For example, when I looked at Figure 12, it was difficult to understand the legend and what the symbols mean without reading the text on a different page. The axes descriptions, legend description, and a summary of the result should always accompany a figure caption.

---

> ### Author Response · Authors · 2024-11-27
>
> Will surely do that! Thanks for the suggestion.

---

> > ### Author Response · Authors · 2024-11-27
> > **Figure Captions**
> >
> > Thank you for your feedback. We have uploaded a revised paper with expanded and self-contained figure captions. We hope this addresses your concerns and improves the overall clarity of the presentation.

---

> > > ### Author Response · Authors · 2024-12-02
> > >
> > > Dear Reviewer kJbZ, as the discussion phase concludes, we’d like to check if we’ve addressed all of your concerns. If anything remains unclear, we’d be happy to provide further clarification.

---

### Note · Authors · 2025-11-09

**Comment:**

We are withdrawing this submission as the conference has concluded.

**Withdrawal Confirmation:**

I have read and agree with the venue's withdrawal policy on behalf of myself and my co-authors.

---

### Meta-Review · Area_Chair_YXXR · 2024-12-18

**Metareview:**

This paper identifies a correlation between the magnitude of perturbations and the level of granularity in the painting process required to optimize classification accuracy. Building on this insight, the proposed Painter-Classifier-Decisioner (PCLD) framework utilizes adversarial training to construct an ensemble of classifiers applied to a sequence of paintings with varying levels of detail. During the rebuttal, the authors provided additional experiments to address reviewers' concerns; however, some issues remain unresolved. The key weakness of this work lies in its lack of evaluation against adaptive attacks, which are critical for demonstrating the robustness of the proposed defenses. Without evidence that the defended model can limit the impact of such attacks compared to an undefended model, the effectiveness of the defense remains uncertain. Furthermore, broader evaluation concerns, including insufficient attack steps, potential gradient obfuscation issues, possible bugs in the attack implementations, and limited extensibility to other perturbation models, undermine the reliability and generalizability of the results. Based on these concerns, we have decided not to accept this work in its current state.

**Additional Comments On Reviewer Discussion:**

Some of the concerns have been successfully address during the rebuttal period but the key concern remains: the limited evaluation against adaptive attacks, which is critical to provide a solid demonstration of the effectiveness of the proposed method.

---

### Decision · Program_Chairs · 2025-01-22

Reject